META-RESEARCH ARTICLE

# What's not in the news headlines or titles of Alzheimer disease articles? #InMice

**Marcia Triunfol** [1]*, **Fabio C. Gouveia** [2]

**1** Research & Toxicology Department, Humane Society International, Lisbon, Portugal, **2** Casa de Oswaldo Cruz, Fiocruz, Rio de Janeiro, Brazil

* mtriunfol@hsi.org

**Data Availability Statement:** We are sending a comprehensive spreadsheet with the raw data for appreciation by the reviewers only. As requested by our data provider Altmetric, these data cannot be

## Abstract

There is increasing scrutiny around how science is communicated to the public. For instance, a Twitter account @justsaysinmice (with 70.4K followers in January 2021) was created to call attention to news headlines that omit that mice, not humans, are the ones for whom the study findings apply. This is the case of many headlines reporting on Alzheimer disease (AD) research. AD is characterized by a degeneration of the human brain, loss of cognition, and behavioral changes, for which no treatment is available. Around 200 rodent models have been developed to study AD, even though AD is an exclusively human condition that does not occur naturally in other species and appears impervious to reproduction in artificial animal models, an information not always disclosed. It is not known what prompts writers of news stories to either omit or acknowledge, in the story's headlines, that the study was done in mice and not in humans. Here, we raised the hypothesis that how science is reported by scientists plays a role on the news reporting. To test this hypothesis, we investigated whether an association exists between articles' titles and news' headlines regarding the omission, or not, of mice. To this end, we analyzed a sample of 623 open-access scientific papers indexed in PubMed in 2018 and 2019 that used mice either as models or as the biological source for experimental studies in AD research. We found a significant association ($p < 0.01$) between articles' titles and news stories' headlines, revealing that when authors omit the species in the paper's title, writers of news stories tend to follow suit. We also found that papers not mentioning mice in their titles are more newsworthy and significantly more tweeted than papers that do. Our study shows that science reporting may affect media reporting and asks for changes in the way we report about findings obtained with animal models used to study human diseases.

## Introduction

Scientists have for some time voiced concern that media reporting of scientific findings, especially those related to health and diseases, are often misleading and that journalists frequently make inappropriate inferences regarding causality, using sensational language to describe scientific findings obtained in biomedical studies [1–3]. In March 2019, James Heathers, of Northwestern University, launched the Twitter account @justsaysinmice (with 70.4K followers

shared with readers due to copy right potential infringements of news headlines.

**Funding:** The authors received no specific funding for this work.

**Competing interests:** The authors have declared that no competing interests exist.

**Abbreviations:** Aβ, amyloid-β; AD, Alzheimer disease; ARRIVE, Animal Research: Reporting of In Vivo Experiments; NSI, nonscientific impact.

in January 2021) to call attention to news stories with headlines omitting that the new medical findings they report on are based on research using mice, not humans. This situation, and the use of exaggerated language, is frequently seen in news reporting of health research, and Alzheimer disease (AD) is no exception [4].

AD is characterized by a degeneration of the brain, loss of cognition, and behavioral changes. The main histopathological cerebral hallmarks of AD are extracellular deposits of amyloid-β (Aβ) plaques and the presence of tau proteins in intracellular neurofibrillary tangles. However, questions abound as to how these 2 proteins orchestrate the onset and progression of AD in the human brain and whether there are other important unknown players involved in disease development and progression. It has been shown that while genetics do play a role in AD, lifestyle and other unknown factors seem to impact an individual's chances of developing AD [5].

Currently, there are no drugs available that can either stop or slow the progress of AD, and the drugs approved by regulatory agencies to be used in AD patients (donepezil, rivastigmine, galantamine, memantine, and a combination of memantine plus donepezil) only treat some of the symptoms of this devastating disease in a limited number of patients [6]. Nevertheless, in 2019 alone, there were 132 agents in clinical trials for the treatment of AD [7]. This dearth of efficacious treatment options is the result of decades of failing clinical trials, in which approximately 99.6% of the drugs previously tested for safety and efficacy in animals were either ineffective or associated with severe side effects when given to AD patients [8]. The poor predictive value of animal models used to study AD, and other human diseases, has resulted in a loss of millions of dollars [9] and yet, drug development success rates are below 5% for most diseases [10].

The first transgenic mouse models of AD were reported in 1995 [11] with the PDAPP model, which was followed by the Tg2576 [12] and the APP23 mouse models [13]. According to ALZFORUM (www.alzforum.org), a comprehensive website that provides information on AD and includes an annotated database of rodent models of neurodegenerative diseases, there are currently 194 models between mouse and rat to study AD. A variety of different approaches have been applied in the attempt to create an animal model that mimics the many characteristics and the cognitive deficiencies seen in AD patients. These approaches include nongenetically modified models in which brain lesions are created in the animal counterpart area believed to be responsible for cognition or by giving injections of Aβ directly into the brain of the animal with the aim to produce some level of cognitive impairment. All these approaches create models with very different pathophysiological and clinical characteristics from the human disease. A number of "humanized" strains of transgenic mice carrying AD mutations have also been created [14]. Each of these models differ from one another, and none exhibits all the main features associated with AD in humans. Even though a number of animals may show some features found in AD patients, e.g., the buildup of amyloid plaques, none presents dementia as recognized in humans. Therefore, it seems correct to state that AD is an exclusively human condition that does not occur naturally in other species and appears impervious to reproduction in artificial animal models.

It is not known what prompts writers of news stories to either omit or acknowledge, in the story's headlines reporting on AD research, that the study was done in mice. To better understand how writers chose their news headlines in this regard, we raised the hypothesis that they are influenced by article's titles. To test this hypothesis, we investigated if research papers whose findings apply to mice, and not to humans, but which omit this caveat in their titles, generate significantly more news stories with headlines that likewise omit mice, if compared to research papers with titles that do mention mice. We also investigated if papers omitting mice in their titles generate more news or tweets. To this end, we analyzed a sample of 623 open-access papers indexed in PubMed in 2018 and 2019 that used mice (the most popular animal

model to study AD) either as models or as a biological source for experimental research in AD—an exclusively human condition that has become an important public health issue affecting millions of people worldwide.

Our findings support our hypothesis that a strong association exists between news stories' headlines and research papers' titles regarding the omission, or not, of mice. News stories' headlines that omit mice as the main study subject may mislead the public regarding the actual state of affairs in AD research, while they may raise false hopes for patients and their families.

## Results

### The sampling process

Our sample of 623 papers was composed of 2 groups that we named "declarative" and "nondeclarative." The declarative group included papers that declared in their titles that mice were the main study's species ($N = 405$), while the nondeclarative group included papers in which mice were omitted in the paper's title ($N = 218$). Using Altmetric Explorer (a web-based platform that produces a report on digital attention data for research papers, including news stories, social media, and citations), we collected and analyzed the headlines of the digital news stories generated for each group of research papers in our sample and determined whether or not the news headline mentioned mice. Fig 1 shows all steps taken to obtain the groups of papers.

### Papers in both groups generate news stories

For the 405 and 218 papers (S1 and S2 Tables) in the declarative and nondeclarative groups, respectively, Altmetric Explorer tracked 382 (94.3%) papers in the declarative group and 212 (97.2%) papers in the nondeclarative group. Of the 382 tracked papers in the declarative group, 295 (77.2%) generated at least 1 news story (a total of 887 news stories or 3.0 stories per paper on average). From the 212 papers of the nondeclarative group tracked by Altmetric Explorer, 173 (81.6%) generated news stories (a total of 682 news stories or 3.9 stories per paper on average; Fig 1). Headlines that were not in English were translated using Google Translate. Because we were interested in detecting specific words in the title, namely mouse, mice, rodents, murine, or animal—see Materials and methods for details—we did not need precise translations. The first question we raised was "is there a significant difference in the number of papers that generated news stories between groups?" We found that that there is no significant difference between groups ($p = 0.21$; Fig 2).

### Nondeclarative papers are more newsworthy

We next asked whether there was any difference between groups for the number of news stories that each group of papers generated. We found that nondeclarative papers generated significantly more news proportionally (3.9 versus 3.0 news per paper or 31.1% more news; Fig 1) than did the papers in the declarative group ($p = 0.012$; Fig 2).

Of the 887 news stories generated by declarative papers, 25 did not have a headline, resulting in a sample of 862 (97.2%) news stories. Of the 682 news stories reporting on papers in the nondeclarative group, 9 did not have any headline. The final number of news stories generated by nondeclarative papers that we worked with was 673 (98.7%; Fig 2). Of note is the fact that the number of papers originating news stories in either group remained the same.

### News writers tend to follow articles' authors on omitting, or not, mice

Next, we examined the headlines of the news stories generated from articles in each group. We found that of 862 news stories reporting on papers in the declarative group, 398 (46.2%) were

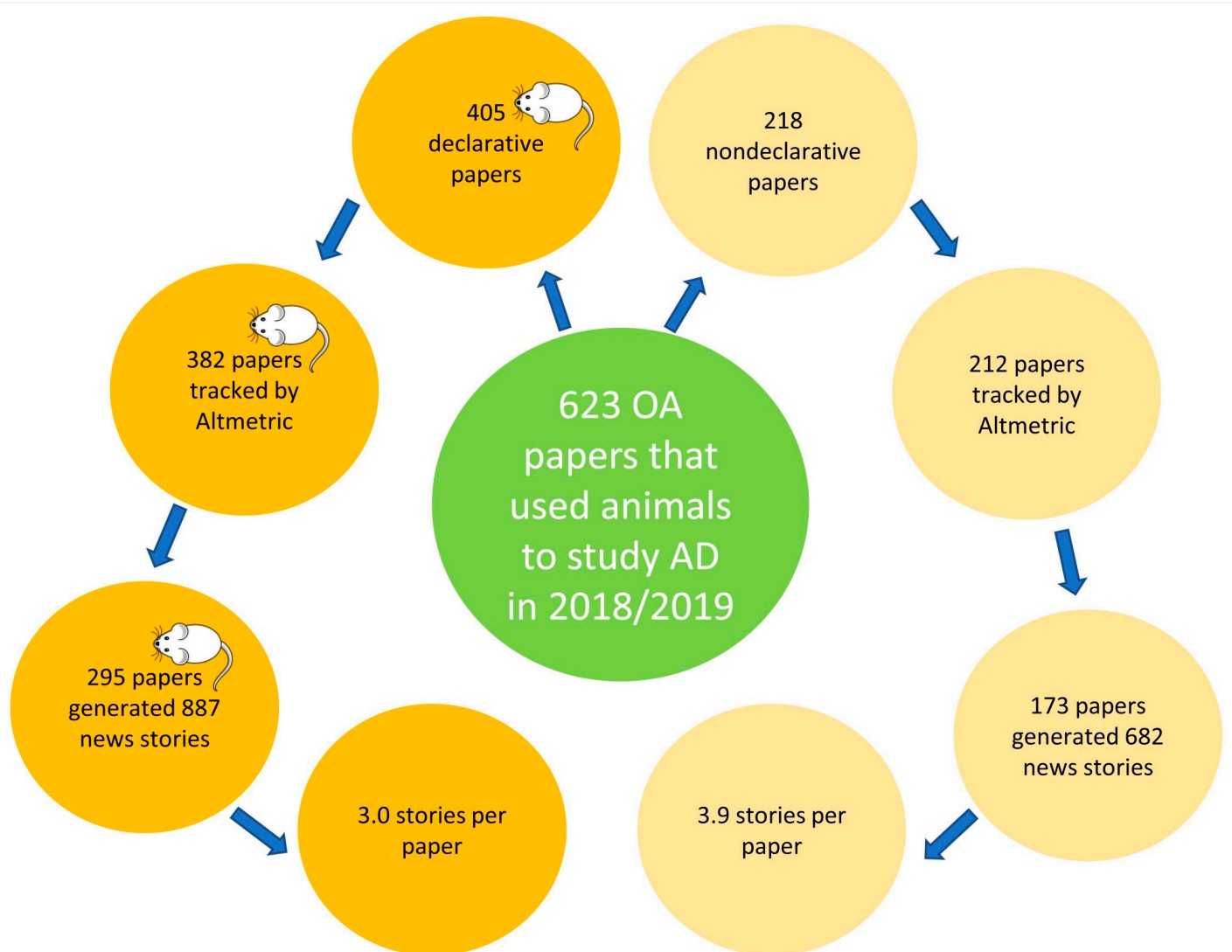

**Fig 1. Sampling process.** Sampling process leading to the groups of papers analyzed in this study. AD, Alzheimer disease; OA, open access.

declarative as well, meaning that 398 news stories have headlines that mention either mice or another qualifying word we considered (see Materials and methods for a list of qualifying words). However, of the 673 news stories reporting on papers in the nondeclarative group, only 70 (10.4%) were declarative, a difference that is highly significant (46.2% versus 10.4%; $p < 0.01$; Fig 2). This finding indicates that when authors mention mice in the paper's title, writers tend to follow suit when crafting the headlines of their news stories.

Note in Fig 2 that only 10.4% (70) of the news stories reporting on papers in the nondeclarative group mention mice in their headlines, compared with 46.2% (398) for news stories that mention mice reporting on papers in the declarative group. Also, 70 declarative news stories were generated from 18 out of 173 papers (10.4%) that generated news stories in the group of nondeclarative papers, while in the group of declarative papers, 398 declarative news stories were generated from 284 papers out of 295 papers (96.3%) that generated news stories. Importantly, of the 1,535 news stories generated from research papers in both groups, only 468 (30.5%) were declarative, while 1,067 (69.5%) omitted mice from their headlines.

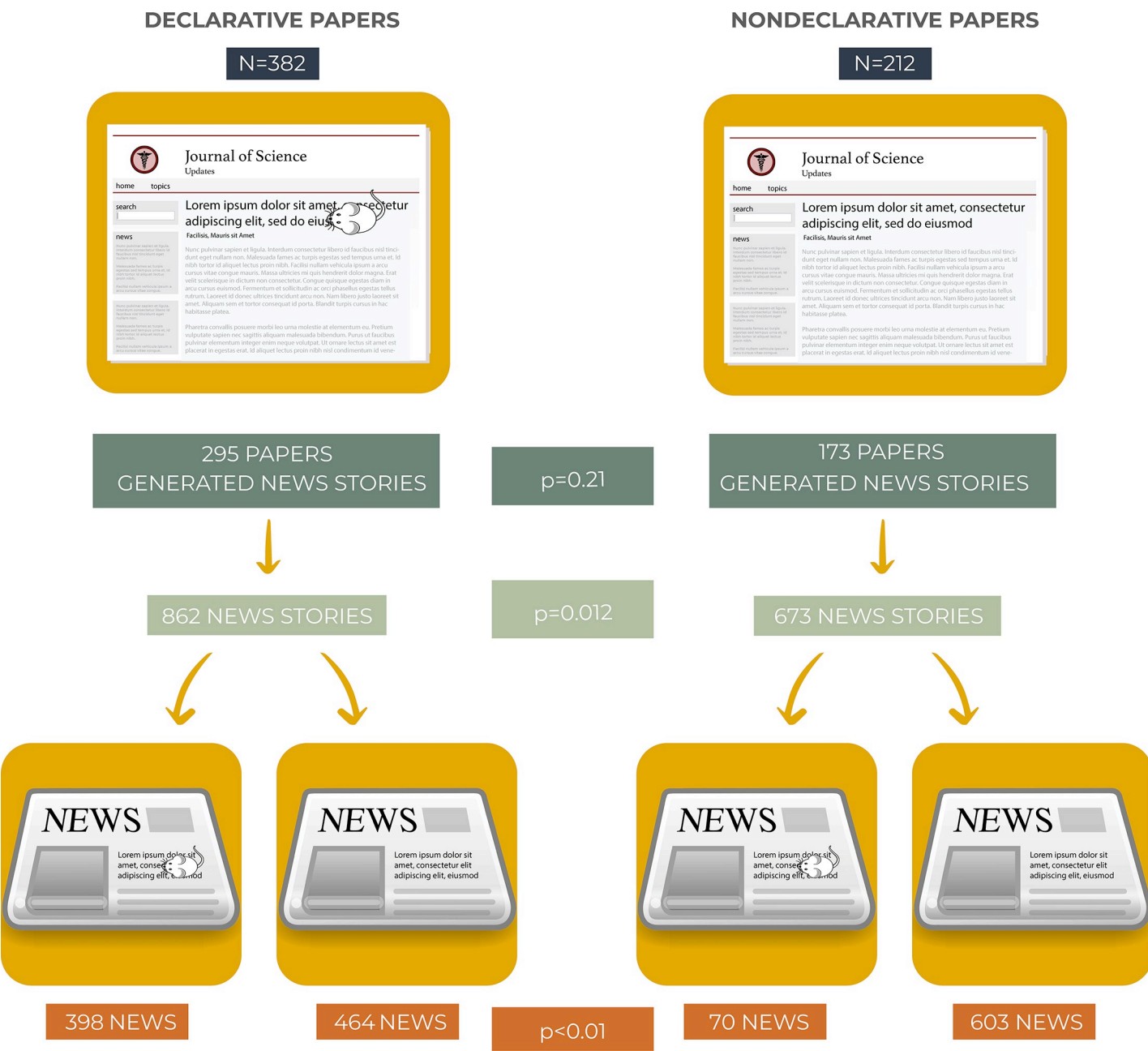

**Fig 2. Significant association between scientific papers' titles and news stories' headlines.** Number of declarative and nondeclarative papers refer to papers tracked by Altmetric Explorer. Number of news stories refer to stories with titles. No statistical difference was found for the number of papers that generated stories in each group ($p = 0.21$), while a significant difference for the number of stories generated by each group of papers was observed ($p = 0.012$). More important, a strong association was found between the type of title in the scientific article (whether or not declarative) and the type of headline in the corresponding news story ($p < 0.01$). Note the drawing of a little mouse in the title and headline of the images representing declarative groups.

News stories posted online often reproduce the original title of the research paper [15]; thus, we examined whether the association we observed between the research paper's title and the news story's headline was a function of research papers' titles being copied into news stories headlines. To answer this question, we first asked if there was a difference between the 2 groups of papers in the number of news stories in which the headline was a verbatim copy of

the paper's title. While in the group of news stories generated from nondeclarative papers 157 headlines were verbatim copies of the research paper's title (157 out of 673 news stories, 23.3%), in the group of news stories generated from declarative papers, this number was 283 (283 out of 862 news stories, 32.8%), a difference that is highly significant ($p < 0.01$). This finding could suggest that titles of declarative papers tend to be more frequently copied into news stories' headlines than titles of nondeclarative papers. However, a closer look at the outlets doing verbatim copies of research papers' titles revealed that a section in the ALZFORUM website called Papers, in which papers in AD are featured and their full references, including their titles, are transcribed, was virtually the only vehicle doing verbatim copies of papers' titles. Also, the apparent significant difference we obtained between the 2 groups is in reality a function of the number of papers that a single vehicle, ALZFORUM, posts on its website.

In any case, we excluded all news stories with headlines that were verbatim copies of their corresponding papers' titles, resulting in 579 original headlines reporting on papers in the declarative group and 516 original headlines for papers in the nondeclarative group. We next tested whether a strong correlation between the type of paper's title (whether or not declarative) and the news story's headlines remained. We still found a highly significant difference between groups ($p < 0.01$; Fig 3).

In the attempt to explain the association between articles' titles and news' headlines, we raised the hypothesis that press releases in EurekAlert! (the main repository for science-related press releases) could influence writers on their news stories. However, we observed that a similar pattern for news stories is also found for EurekAlert! press releases, in which declarative papers generated more declarative releases (5 declarative releases out of 22 releases) if compared to nondeclarative papers (1 in 19). We next searched for press releases for any of the 70 articles in the nondeclarative group of papers that generated news stories that had mice in their headlines. We found that only 1 EurekAlert! press release was produced for a single paper among all the 70 nondeclarative research papers. Thus, press releases do not explain the association we found.

We then asked if there was any particularity in these 70 research papers that could have driven writers to write headlines that mentioned mice. Notably, these research papers generated news stories omitting mice in their headlines as well as others acknowledging it, indicating that nothing particular in these papers could be driving writers to choose one way or the other. An initial analysis of the news outlets did not reveal a pattern that could indicate that omitting, or not, mice in the headline was an editorial decision of any kind.

## Journal choice does not explain our findings

We next asked whether any of the journals in which these papers were published required that authors informed in the article's title the main species used in the study. This requirement could explain why some authors acknowledged the use of mice in the paper's title while others did not. To find this information, we referred to the guidelines of the journals. All 468 papers in both groups that generated news stories (295 + 173) were published in 157 different journals. Of this list, one journal (*Cellular Physiology and Biochemistry*) is no longer in business due to concerns about the integrity of papers published in the journal in 2017 and 2018. In our samples, we only had 2 papers published in this journal, 1 in each group. Of the 156 journals analyzed, only 8 (5.1%) require that the species used in the study is informed in the paper's title (S3 Table).

Next, we checked whether there was any article in the nondeclarative group that was published in any of the 8 journals requiring the study's species to be acknowledged in the title. Surprisingly, we found that 11 (6.4%) articles were published in such journals, indicating that

## DECLARATIVE PAPERS

N=382

579 ORIGINAL HEADLINES

115 STORIES    464 STORIES    p<0.01

## NONDECLARATIVE PAPERS

N=212

516 ORIGINAL HEADLINES

70 STORIES    446 STORIES

**Fig 3. Significant association between original headlines and papers' titles.** Number of original news stories' headlines and their association with research papers' titles. Of the 579 original headlines generated from declarative papers, 115 (19.86%) acknowledged mice in their headlines, while only 70 original news stories' headlines (13.57%) generated from nondeclarative papers did the same, a difference that is highly significant ($p < 0.01$). Note the drawing of a little mouse in the title and headline of the images representing declarative groups. Two news headlines were truncated and we did not find their full versions online. Because the truncated parts were similar to the articles' titles, they were considered verbatim copies.

even though these journals require that authors inform the study's species in the paper's title, they do not enforce authors to do it. As for articles in the declarative group, we found that only 7.8% papers (23 out of 295 papers) were published in journals that requested the title of the paper to inform the species used in the study. This finding indicates that for most papers in this group (272 papers or 92.2%), acknowledging the studies' species in the paper's title was most likely an authorial decision rather than one imposed by journals' requirements.

Another constraint that may have played a role in authors mentioning mice, or not, in the paper's title could be word or character number limitations set by the journal guidelines. To investigate this possibility, we again referred to the journals' guidelines. We found that of the 156 journals we examined, only 39 (25.0%) limited the number of either characters or words allowed in the paper's title (S3 Table).

**Fig 4. Significant association between research papers' titles and tweet counts.** We found a significant association between the research paper's title (whether or not declarative) and the number of tweets the research paper generates ($p < 0.01$). Papers' titles that omit animals are more often tweeted than titles that openly acknowledge that the study was done in animals. Note the drawing of a little mouse in the title of the image representing the declarative group of papers.

We then found that 63 papers out of 173 (36.4%) in the nondeclarative group were published in journals that limit the number of either words or characters in the title, while in the declarative group of 295 papers, this number was 90 (30.5%), a difference that is not significant.

### Articles' titles omitting mice are more often tweeted

We then compared the number of tweets in each group of papers and found a significant difference for the mean number of tweets per paper in each group. While in the declarative group of research papers each paper generated 9.7 tweets (mean number), in the nondeclarative group, each paper generated 18.8 tweets ($p < 0.01$), indicating that articles' titles that omit mice are more often tweeted than articles' titles that openly acknowledge that the study findings apply to mice. Overall, papers in the nondeclarative group generated 2,632 tweets, while papers in the declarative group generated 1,902 tweets (Fig 4).

### Discussion

Here, we investigated the hypothesis that studies using mice as the main research subject, but which omit this information in their articles' titles, generate significantly more news stories with headlines that likewise omit mice. Our data support our hypothesis that news writers are influenced by article's titles regarding the omission, or not, of mice in their headlines.

Titles of scientific papers provide short and crucial information for paper retrieval and storage while aiming to raise the reader's interest. Also, in biomedical research, most titles highlight some aspect of the study's findings, as opposed to titles that limit to inform what was

investigated but offer no information on the results obtained. The former is believed to better attract the reader interest, and, nowadays, is the preferred style adopted. Thus, most titles do carry information about the study results.

Previous studies have shown that the title can have a significant impact on how frequently the paper is cited by others [15]. Of note is the fact that in biomedical studies, the species used is assumed to have been humans, unless the authors inform otherwise. For this reason, most books and guidelines on how to write a title of a scientific paper advise authors that the species used in the study should be informed in the article's title, when not humans [16–18]. As recommended by Ushma S. Neill in 2007 [19], then the executive editor of the high-impact *Journal of Clinical Investigation*, "try to get the species studied into the title; this is sometimes complicated if multiple species are used, but it allows readers to immediately put the work in its appropriate context."

The main argument raised to justify the use of animals as models for human diseases is to advance knowledge about certain human conditions. As already mentioned in the introduction, AD is a condition that only humans develop, and which does not occur naturally, nor can it be experimentally induced in other species. As Dr. Susan Molchan, a geriatric psychiatry, and former NIH researcher and FDA reviewer has once said, "they've cured *mouseheimer's* disease I don't know how many times now," when referring to scientists working with mice to understand AD [20]. Therefore, AD animal models should not be considered as full and valid representations of the human condition. This is a caveat that any study using animal models should inform up front, preferentially in its paper's title. However, in our sample of 623 mice studies on AD, 34.9% of the papers (218 out of 623) omitted mice in their titles (Fig 1).

As articulated by Daniel Dor in a study entitled "On Newspaper Headlines as Relevance Optimizers" [21] "traditionally, newspaper headlines have been functionally characterized as short, telegram-like summaries of their news items. This is especially true with respect to news headlines." Headlines play an important role in communication, and they may be the only piece of information that is actually read [2,21]. Indeed, according to a survey done by the Pew Research Center on Science News and Information in 2017 [22], only half of social media users click through science news stories. Of these, 10% do it often, 43% do it sometimes, and the remaining do it hardly, never, or simply do not see science news stories. Therefore, those who do not click through the story are left with the impression caused by the science news headlines only. Also, a study done on social clicks that used multiple data collection techniques to analyze Twitter conversations and clicks for URLs from 5 news sources show that most (59%) of the links on news stories shared in social media (Twitter) are never read [23]. Thus, it seems reasonable to conclude that for most online users, the only part of the news stories read by those who share them is the headline.

In this study, we found no significant difference in the number of papers that generated news stories between groups. However, when we compared the 2 groups of papers for the number of news stories each generated, we found that nondeclarative papers generated significantly more news stories than declarative papers. This finding indicates that nondeclarative papers are more "newsworthy," if compared to declarative papers. Also, news stories that report on nondeclarative papers tend to omit in their headline that the study's finding applies to mice and not to humans, revealing that the title of the research paper influences the news headline's writer regarding omitting, or not, mice. The vast majority (69.5%; 1,067 stories out of 1,535; Fig 2) of the news stories generated from papers in both groups omit mice in their headlines. Often, these headlines overclaim the scientific finding and can lead to misperceptions by the general public. The use of misleading language in news reporting on AD research is not a rare event [4] and has been observed in other diseases such as cystic fibrosis [24] and multiple sclerosis [25].

It has been shown that information that sounds useful to others and which carry a positive and emotionally charged frame that evokes hope has higher chances to be replicated and shared by others [26]. For instance, news headlines such as (i) a common dietary supplement could be the answer to fighting AD; (ii) a diet high in salt causes dementia and other brain problems (translated from Spanish); or (iii) a long-standing antibiotic offers a new path against AD, among others, may leave the public with the impression that these findings are useful and offer hope to AD patients, when in reality they apply to mice, unless new scientific evidence is produced.

These headlines can be considered misleading, which, according to previous studies, can lead to misconceptions and misinformation. Even if readers read the full article, it may not be enough to deconstruct the impression left by a misleading headline [12]. Indeed, a survey done by Harvard School of Public Health and Alzheimer Europe that heard 2,678 individuals in 5 countries (France, Germany, Poland, Spain, and the United States) found that roughly half of the individuals believed that an effective treatment for AD was available at the time or would be available in the next 5 years. The survey was done in 2011 [27]. The dissatisfaction with health reporting has been recognized by journalists, outlet media, and by media-reporting organizations. This sentiment has prompted the creation of groups that oversight news coverage, such as HealthNewsReview.org, and the production of guidelines on how to report health news, by media outlets. One of the articles by HealthNewsReview.org entitled "Why you should be cautious of health claims based on animal and lab studies" aims at helping journalists understand the many caveats of animal studies and how better report on such studies [20].

Another example is the guidelines created for The Age, The Sydney Morning Herald, Brisbane Times, and WAtoday, in Australia, that among many topics explicitly describes that journalists ". . . will treat research based on animal studies with caution, preferring to focus on human trials, and making clear that the results may not translate to human trials" [28]. Similarly, Science Media Centre has produced a guideline that describes 10 best practices for reporting science and health stories that includes "headlines should not mislead the reader about a story's contents and quotation marks should not be used to dress up overstatement" [29]. Nevertheless, the survey done by the Pew Research Center on Science News and Information [22] revealed that 43% of US adults say that the news media are "too quick to report research findings that may not hold up." A smaller number of adults (30%) sees as a problem the fact that media oversimplifies scientific findings. Interestingly, 4 in 10 Americans (40%) believe that researchers also play a role on how science news are covered, an impression supported by our findings.

But what objective forces would be driving authors to omit, or not, mice in their research papers' titles? We considered 2 main hypotheses: a requirement, as defined by the journal guidelines, for the study species to be acknowledged in the paper's title and/or a constraint imposed by the journal on the number of words or characters allowed in the paper's title. Neither hypothesis was supported by our data, and thus mentioning or not mice in the title seems to be an authorial decision, for which peer reviewers and journal editors may also play a role, and may reflect particular, and subjective, views.

In 2001, world-renowned evolutionary biologist Svante Pääbo had predicted that the sequencing of the human genome would reveal that our species shares a genetic scaffold with all species on Earth. For mice in particular, the number of genes and the general structure of the genome were expected to be similar to that of humans, highlighting our similarities [30]. Thus, it is possible that some researchers truly see animals, and mice in particular, as smaller and less complex creatures resembling humans, while overlooking the many differences between species at physiological, metabolic, and behavioral levels. Thus, mice may be considered by some as the unspoken norm, and its use in biomedical research is not seen as a caveat that requires further explanation.

Alternatively, a growing body of studies has called into question the scientific value of using animals in biomedical research, thus omitting animals in the study's title may be a reaction (consciously or not) to this trend [31–35]. It is also possible that some studies' authors perceive that omitting mice in the title would give them better chances of grabbing the journal's editor attention or provide higher visibility, which could yield more citations [36]. Indeed, we found that research papers omitting mice in titles generate significantly more news and more tweets than papers that do mention mice (Figs 2 and 4). Thus, omitting mice in the paper's title significantly improves the paper nonscientific impact (NSI), which has been defined as "the amount of attention given to scientific research by nonscientists in mainstream news outlets, online blogs, and/or social media" [37].

In a recent study that investigated whether an association exists between NSI and the number of citations a given study receives, authors found that research papers that receive more attention in nonscientific media are cited more by other peer-reviewed studies [37]. Thus, it is possible that papers receiving more media attention are and will be more cited. However, because here we considered papers published in 2018 and 2019, we would need more time for citations to accrue before this analysis can be done.

Spin is a concept more often associated with propaganda and journalism, used to influence public opinion, but which has also been found in research articles. Spin may be introduced by many means of p-hacking [38], manipulation of figures [39], or selective reporting of outcomes [40] and may have consequences on researchers' interpretation of the study findings or clinical decision-making. Thus, omitting mice from the paper's title may be considered a new class of spin [41] that we propose to classify as "misrepresentation" since the title of the paper is not a short and accurate representation of the full study but rather a misleading brief account that omits an important caveat: the fact that the study was done in mice. Due to the higher NSI of these papers and the fact that most online news stories and most tweets (which are often a mere copy of the work's title [42]) are not read through, plus the fact that most titles in biomedical research summarize the study findings, this omission may lead to a misunderstanding of the actual implications of the study's results for human health and its translatable potential to people suffering with AD, contributing to a wave of science misinformation.

Our study has limitations. The articles analyzed in either group constitutes a convenience sample that includes only research articles that are available as open access. This sample represents roughly half of all papers that used mice to study AD during 2 periods (2018 and 2019). We should also add that the results found here apply exclusively for the lists of articles on AD research we investigated and cannot be expanded to other areas without careful analysis. Also, we did not inspect each article thoroughly to check whether there is any characteristic inherent to the study that may explain the choice for mentioning mice, or not, in the title.

To our knowledge, this is the first study to present scientific evidence that the way science is reported by scientists plays a role on the way science news is reported by journalists. Whatever the underlying reasons for such omissions, it is expected that scientists communicate their findings accurately and at high standards. In fact, a much simpler argument is this: Why hiding in the article's title the species used in the study?

The 2010 NC3Rs Animal Research: Reporting of In Vivo Experiments (ARRIVE) guidelines, developed by the UK National Centre for the 3Rs to improve reporting of studies using animals, did not specifically recommend authors to inform the species, or strain, used in the study in the title of the research article, being this recommendation limited to the article's abstract. The 2020 update to the ARRIVE guidelines [43] does not identify the title of a paper as an important element subjected to review. However, we believe the findings presented here provide grounds for further amendment of the ARRIVE guidelines, together with journal publication policy, to give special attention to the article's title and require the titles of papers

describing experimental studies to identify the species and/or tissue sources used in the research.

We expect that our recommendation for amendment to the ARRIVE guidelines will have an impact on the headlines of news stories and then we should start seeing more news stories that, as urged by James Heathers and his more than 70,000 Twitter followers, "just says in mice" in their headlines.

## Materials and methods

### Study design

We retrieved papers published in 2018 and 2019 and which were indexed in PubMed, which used mice to study AD. Papers published in 2018 were retrieved in June of 2020, and papers published in 2019 were retrieved in December of the same year. We were interested in retrieving 2 different groups of papers: The declarative group would include papers that mention mice in the title, while the nondeclarative group would be formed by papers that did not mention mice in the title. We selected research papers that were published in English in open-access journals and which were not comments or reviews.

We used the PubMed ID of each paper identified in our search to retrieve the matching research outputs using Altmetric Explorer. We used the Altmetric Explorer data to determine the number of news stories that each group of papers generated. We also analyzed the number of tweets that each group of paper generated and the mean number of tweets per paper in each group.

### Data extraction from PubMed

Below are the 2 strings we built to extract the data:

String to retrieve papers published in 2018 in the declarative group:

("mice"[All Fields] AND "alzheimer disease"[MeSH Terms] AND 2018:2018[PDAT] AND "English"[Language] AND "journal article"[Publication Type] AND "loattrfree full text" [Filter]) NOT ("review"[Publication Type] OR "systematic review"[Publication Type] OR "patient"[All Fields] OR "patients"[All Fields]) AND ("mice"[Title] OR "mouse"[Title] OR "rodent"[Title] OR "murine"[Title] OR "animal"[Title])

A similar string was used to retrieve papers published in 2019. Merging the results obtained with each query and removing overlaps (in some cases, PubMed retrieved the same paper twice in the 2018 and 2019 search results), we ended up with a sample of 405 papers. However, for our analysis, we only considered papers that were tracked by Altmetric Explorer. Thus, our final number of declarative papers was 382 (Fig 1). To be included in the declarative group, a paper had to have one of the following words in its title: mice, mouse, rodent, murine, or animal.

String to retrieve papers published in 2018 in the nondeclarative group:

("mice"[All Fields] AND "alzheimer disease"[MeSH Terms] AND 2018:2018[PDAT] AND "English"[Language] AND "journal article"[Publication Type] AND "loattrfree full text" [Filter]) NOT ("review"[Publication Type] OR "systematic review"[Publication Type] OR "patient"[All Fields] OR "patients"[All Fields]) NOT ("mice"[Title] OR "mouse"[Title] OR "rodent"[Title] OR "murine"[Title] OR "animal"[Title])

A similar string was used to retrieve papers published in 2019. This query retrieved 201 papers published in 2018 and 194 in 2019. However, we wanted our group of nondeclarative papers to be limited to papers that used mice only. If a second species of rodent was used in the study, we kept the paper in the group (these were very few papers). Papers that used both mice and humans (ex vivo, patients' samples, or human cells for culture) were removed.

Studies done entirely in silico were also removed. Studies using genetically modified mice carrying human genes were kept. This screening was done by one of the authors only, MT. Our final sample of papers published in 2018 was 108, and, of papers published in 2019, was 117. After removing overlaps, we obtained a sample of 218 nondeclarative papers. Finally, the number of papers tracked by Altmetric Explorer was 212 (Fig 1).

A table with all tracked references in each group is available as Supporting information (S1 and S2 Tables).

## Statistical analysis

We applied chi-squared test at a $p < 0.01$ to investigate whether the differences observed between groups could be considered significant.

## Supporting information

**S1 Table. List of declarative papers ($N = 405$).**
(PDF)

**S2 Table. List of nondeclarative papers ($N = 218$).**
(PDF)

**S3 Table. List of journals and their guidelines for papers' titles ($N = 156$).**
(PDF)

## Acknowledgments

The authors appreciate the inputs provided by Catherine Willett, Bianca Marigliani, Lindsay Marshall, Troy Seidle, Hanna Stuart, and Helder Constantino. Special thanks to Angela Paes for reviewing the statistical analysis. The authors also would like to thank Altmetric for providing access to Altmetric Explorer for Researchers for the purpose of the present research. Fábio Castro Gouveia would like to thank the National Council for Scientific and Technological Development (http://www.cnpq.br) for grants 315521/2020-1 and 430982/2018-6.

## Author Contributions

**Conceptualization:** Marcia Triunfol.

**Data curation:** Marcia Triunfol, Fabio C. Gouveia.

**Investigation:** Marcia Triunfol, Fabio C. Gouveia.

**Methodology:** Marcia Triunfol, Fabio C. Gouveia.

**Validation:** Marcia Triunfol, Fabio C. Gouveia.

**Writing – original draft:** Marcia Triunfol.

**Writing – review & editing:** Marcia Triunfol, Fabio C. Gouveia.

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
