## [Editor Report · Decision Letter 0]

10 Sep 2020

Dear Dr Triunfol, 

Thank you for submitting your manuscript entitled "What’s not in the Title? The Mice Used in the Study." for consideration as a Short Report by PLOS Biology.

Your manuscript has now been evaluated by the PLOS Biology editorial staff as well as by an academic editor with relevant expertise and I am writing to let you know that we would like to send your submission out for external peer review. However, we would like to consider your manuscript as a Meta-Research Article, rather than as a Short Report. We do think that given that the article is about reporting science, it would fit better as that type of article. Please select Meta-Research Article from the type of article menu when you are ready to complete your submission (see below).

Before we can send your manuscript to reviewers, we need you to complete your submission by providing the metadata that is required for full assessment. To this end, please login to Editorial Manager where you will find the paper in the 'Submissions Needing Revisions' folder on your homepage. Please click 'Revise Submission' from the Action Links and complete all additional questions in the submission questionnaire.

Please re-submit your manuscript within two working days, i.e. by Sep 12 2020 11:59PM.

Kind regards,

Ines

--

Ines Alvarez-Garcia, PhD

Senior Editor

PLOS Biology

---

## [Decision Letter · Decision Letter 1]

31 Oct 2020

Dear Dr Triunfol,

Thank you very much for submitting your manuscript "What’s not in the Title? The Mice Used in the Study." for consideration as a Meta-Research Article at PLOS Biology. Thank you also for your patience as we completed our editorial process, and please accept my apologies for the delay in providing you with our decision. Your manuscript has been evaluated by the PLOS Biology editors, an Academic Editor with relevant expertise, and by two independent reviewers.

As you will see, both reviewers find the conclusions of the study interesting and think it is worth pursuing the manuscript for publication, however they also raise several issues that need to be addressed. The main concern is that it is not clear what is actually measured in the study and that this is crucial to be able to draw solid conclusions. In addition, Reviewer 1 thinks you don’t test the quality of the reporting in the articles beyond the title or explore other factors that might influence it – like journal guidelines, for example. After discussing the reviews with the academic editor, we agree that all the comments should be addressed and that the clarifications about the methods are essential. We also think that conclusions need to be more focused on the actual findings and their limitations, and avoid speculation. 

In light of the reviews (attached below), we will not be able to accept the current version of the manuscript, but we would welcome re-submission of a much-revised version that takes into account the reviewers' comments. We cannot make any decision about publication until we have seen the revised manuscript and your response to the reviewers' comments. Your revised manuscript is also likely to be sent for further evaluation by the reviewers.

We expect to receive your revised manuscript within 3 months. 

**IMPORTANT - SUBMITTING YOUR REVISION**

*Re-submission Checklist*

*Published Peer Review*

*PLOS Data Policy*

Sincerely,

Ines

--

Ines Alvarez-Garcia, PhD,

Senior Editor,

ialvarez-garcia@plos.org,

PLOS Biology

Reviewers’ comments

Rev. 1: Kieron Rooney – this reviewer has waived anonymity

The general premise of this article is that the media reporting of studies is misleading / sensationalised on the basis of titles of either studies and/or subsequent news outlet reports not specifying animal species (mice specifically in this case report).

A combined PUBMED and altmetric search for articles specific to mouse models of Alzheimer Disease in one year (2018) was then utilised to test the influence of declaring species (or not) on title in a paper subsequently impacting the reporting in lay media outlets.

Upon finding some significant interactions between the study title influencing reporting in lay media and twitter reporting, the discussion focusses on building a platform for change in regulation of study titles - predominantly through revision of ARRIVE guidelines. The platform built infers (particularly on page 6, lines 65 - page 7, lines 85 AND page 5 lines 18 - page 6, lines 30) that there is some intention in omitting species from title by researchers.

My major concern with this manuscript, is that the discussion is a large over reach for the methods and data presented and arguably the authors could be accused of the same intention to sensationalise their outcomes that the AD community of 2018 are accused of.

At best, this is a thought provoking prod at the quality of titles used in a small subset of studies in a niche area of animal models. The data is no doubt interesting but is more hypothesis generating than testing. For example, the assumption is that the title is all that is important, and fair enough, the authors provide a reference to suggest the title is all that some people read. However, there was no attempt by the authors to further test the quality of media reporting by actually reading and providing an analysis of the content of the article. The bigger question to ask here, in testing the need for better regulation on titles in studies is whether or not the totality of reporting is omitting the use of animals. Many readers of lay articles could feasibly be hooked by the title, find very quickly that the article is reporting a study in animals and either read on or not. In the way that the current manuscript is presented I find the analysis having stopped solely at title a significant flaw in the novelty and significance of the manuscript. That is to say, I had not been convinced by the authors of this manuscript that the title, independent of a complete article is a significant issue to be concerned about.

Given the small number of studies identified, this manuscript could have been significantly enhanced by merely emailing the authors of the identified articles and asking them why they omitted or declared the animal species. Whether or not all replied is not an issue, but atleast some more considered primary data could have been used to support or refute the speculation provided in the current discussion.

There is also no consideration on the "why" a primary study gets reported in the media, and this is an important question to ask given the authors make inferences that animal species is omitted to gain more hits and attention. By raising this point, the authors identify a significant limitation - there is no consideration of why or how a specific primary study reaches lay media. Knowing this is important information given that certain institutions and authors are likely to predominate lay media on the basis of reputation and output, and as such a common culture could be influencing the study data here, beyond the limited considerations (such as spin) presented here.

Further, there was no consideration in the discussion provided to the contradictory result in which lay media declared animal species (n=38) despite originating from non-declared studies. Ignoring this outcome is akin to selective reporting. The authors need to be careful of accusing researchers of a malpractice if their approach itself can be questioned. For my read, the presence of this contradictory result (to the authors premise) is an important one to explore and provides evidence that there is more in the translation of a research paper title to the lay media title that needs to be explored. What is the role of the journalist here? Why did some include the animal species when the original study did not? I am not suggesting that the authors have to explore this point as a dataset to present, but there does need to be more adequate representation / discussion of this result in the discussion.

Finally, there is little if any recognition of what / who determines the title of their paper. That is disappointing given that is the cornerstone on which this work has been completed. We are given, as readers some speculation of spin and intentional deception by authors to gain more significance for their work, but for the most part, title content is determined at the journal level. Authors are provided a word or character limit, some journals will specify species identification is necessary. The authors of this manuscript could have value added to this work by reviewing the author guidelines of the journals included in their analysis to present an interesting discussion regarding whether or not, the title of published papers was adequately peer reviewed against journal requirements, or if journals themselves are the better target for regulation. Currently the authors conclude that an outcome of this work is revision of the ARRIVE guidelines to include a specification of species in title. However in the most recent ARRIVE 2.0 guideline, "title" is not identified as an item for review having been deleted from the original 2010 ARRIVE guidelines. A stronger argument could be built with a more considered investigation of the accuracy of reporting by the studies in this manuscript against the journal author guidelines.

Otherwise, the article was an interesting exploration that could inspire more investigation and as such could be an important contribution to the global debate on accurate reporting of animal use in research. The authors may want to consider some of the points raised above however, either in preparation for final acceptance or in the design of a follow-up study.

Editorial correction:

One typo (well done), Discussion, Page 5, line 97 "exits" I believe should be "exists"

Rev. 2: Quinn Grundy – this reviewer has waived anonymity

The authors conduct a meta-research study to examining reporting practices in research for Alzheimer's Disease conducted in mice models. Specifically, they examine whether the mouse model is specified in the title and whether this reporting is consistent between published papers and news reports on the study. The study relies on a convenience sample of open access papers published during 2018.

This is an interesting study and highlights important issues for science communication. I think the paper could be strengthened if these implications for science communication more generally were identified at the outset (ie the consequences of this specific type of 'spin'). The paper also needs some revisions in terms of reporting and particularly, the outcome reporting.

Major comments:

In stating the research question, throughout the paper, including in the Abstract, Author Summary and Introduction, it is not completely clear what the authors actually measured. Although the authors state they measure an association, the outcome is unclear and seems to suggest they measured concordance/discordance. Simply stating the researchers investigated "whether a relationship exists" is vague; as the papers and news articles report on the same study, of course they are related. At the end of the introduction the authors state, "Our main interest was in determining whether the news headlines and the research papers' titles they refer to follow the same pattern regarding the omission, or not, of mice." This should be reflected throughout the paper and might better be described as consistency or concordance in reporting. Similarly, when reporting the findings, be consistent with your language and describe this as concordant reporting or consistent reporting rather than "media perception" (line 105), which is something entirely different.

I found the Results section difficult to follow. It was often difficult to understand what exactly was measured. Often I didn't understand the distinction between one group of results and the next as it was difficult to ascertain what was being compared. I think it would help to consistently report denominators and proportions alongside numerators and to re-organize the section so that results for the full sample are clearly reported, followed by sub-group analyses (I couldn't quite tell if this occurred), sensitivity analyses (excluding titles copied verbatim), and then secondary outcome (tweets). For example, I couldn't tell what the findings on line 150-151 referred to ("Of the 853 total news pieces generated from research papers in both groups, only 229 (26.8%) were declarative, while 624 (73.2%) omitted this fact from their headlines."

The Materials and Methods section could be strengthened. I would suggest the use of sub-headings including: Study design; Research questions and hypotheses; Data extraction; and Statistical Analysis. Key aspects that are missing are description of the outcome measures. This is a weakness throughout the paper and should be very clearly stated in the Methods, with corresponding analyses detailed. Further details re: who did the screening and whether it was done in duplicate would be useful. A Prisma-type flow diagram would be very useful to summarize the sampling and reasons for exclusion.

Specific comments:

Introduction, first sentence: Not just scientists are concerned and in fact, scientists sometimes the source of hype/spin

By line 90-91 on page 2, I came to understand your point about the appropriateness of mouse models for understanding Alzheimer's disease, however, this understanding would have been more helpful to have up around line 52-53. I would suggest stating this main point (ie Alzheimer's Disease is a human condition; scientists have created mouse models to attempt to study the disease; but, animal models have poor predictive value) following the research aim/question and then, walking the reader through the explanations. I am not a biologist, so the explanations were most helpful.

The sentences in the abstract, author summary and the introduction are sometimes quite long and could be broken up for readability (e.g. first sentence of the Introduction, lines 105-109).

On line 94, what is the significance of the clause "what the studies' authors consider to be AD" - could you explain?

Please avoid all non-standard acronyms including AD, non-GM - please spell out.

Your data availability statement suggests the data are proprietary/copyrighted, but the Methods on 98-100 suggest this platform is open access. Can you explain further how these data were accessed and if public, why the raw data cannot be made public? Further, if they are news headlines, are these data already in the public domain?

Could you avoid using NONDECLAPAPERS and DECLAPAPERS and simply describe the groups as you did in the Introduction? Or come up with more reader-friendly terminology?

You report "a declarative title does not impact media interest in the study." To avoid the suggestion of causality or directionality, please simply state that there was no difference between the groups.

This sentence strikes me as belonging in the Discussion "This finding indicates that when authors openly acknowledge the use of mice, or equivalent qualifying term, in the title of their research article, writers follow the same pattern when crafting the headlines for their news stories."

The Discussion feels a bit untethered from the study's findings at several points. I would suggest re-organizing to clearly summarize the study's key findings. Then, place each finding in the context of the wider literature in each paragraph. The discussion on the appropriateness of animal models for Alzheimer's research is interesting, however, doesn't really reflect the study's findings, so could perhaps be mentioned more succinctly. The literature on spin and the recommendations regarding reporting guidelines seem more relevant, but should be linked to specific findings.

A limitation is that this is a convenience sample, however, I wonder if open access articles are more likely to be picked up by journalists? Is there any evidence for this?

The Figures are engaging, but particularly in Figure 2, I can't understand what the final row of News articles refers to.

---

## [Decision Letter · Decision Letter 2]

23 Mar 2021

Dear Dr Triunfol,

Thank you for submitting your revised Meta-Research Article entitled "What’s not in the Title? The Mice Used to Study Alzheimer's Disease" for publication in PLOS Biology. I have now obtained advice from the two original reviewers and have discussed their comments with the Academic Editor.

The reviews are attached below. You will see that both reviewers think that the manuscript is very much improved, but they also raise a few minor points that need to be addressed. Among them, they would like you to streamline the discussion and the text in general, to make it less redundant in some parts and to improve the flow. Please note that we don't include Authors Summaries in the articles anymore, so please remove it from the manuscript. In addition, we would like to make a suggestion for the title as following:

“#InMice: strong association between omission of animal model information in the title of Alzheimer Disease articles and it absence in the related news headlines”

Based on the reviews, we will probably accept this manuscript for publication, provided you satisfactorily address the remaining points raised by the reviewers. Please also make sure to address the following data and other policy-related requests.

We expect to receive your revised manuscript within two weeks. 

*Published Peer Review History*

*Early Version*

Sincerely,

Ines

--

Ines Alvarez-Garcia, PhD,

Senior Editor,

PLOS Biology

Reviewers’ comments

Rev. 1:

The authors have performed a great amount of work to address the comments raised by myself and the other reviewers. This is highly commendable. I do hope that in keeping with their platform for improved transparency that they adopt the PLOS option to publish the peer review history.

I do still have some minor issues, but I do not believe that these should prevent publication. I hope however that the authors consider in their own reflections that for my read, this paper has still not explained the why the outcome observed occurs. During the review process the authors have preferred not to interrogate further the question they raise themselves in the introduction "However, it is not known what prompts writers of news stories to either omit or knowledge (sic), in the story's headlines" This "why" this present paper raises is a more interesting question to me than the analysis presented, which I still think is more a prod than a comprehensive examination. To this end I would suggest that the authors reflect on the comment here that merely increasing the sample size as was identified in response to my query does not answer the "why does something happen", it simply increases the sample size for the "what happened". However I am content that the "why" is more a future question to ask from this work now and not a priority for the current manuscript as the major issues I had with the overreaching and speculation has been significantly dampened. In dampening the speculation, the need for a stronger mechanism / why something has happened has been reduced and this more comprehensive analysis of the outcome reported is more acceptable.

There are a few minor editorial / typographical errors that stood out and I have listed some below, but in general the new discussion, while more circumspect would benefit from another editorial read over, as there are some very long sentences, separated by commas, that make it somewhat difficult to read, in places.

Line 23: Should "knowledge" be "acknowledge"

Line 106: As for Line 23

Line 294: Should "though" be "through"

Line 304: SHould "that" be "than" (this does occur elsewhere in a few places and the authors should confirm / check throughout)

Line 364: Should "constrain" be "Constraint"

Lines 437 - 441: (principally line 438) it is not just that the ARRIVE 2.0 "still does not identify the title of a paper as an important place to acknowledge the species, or strain, used in the study" The ARRIVE 2.0 don't even stipulate the title as an element to review!

Rev. 2:

Thank you for the opportunity to re-review this manuscript. The additional analyses are interesting and I think address some of the questions about the validity of the associations measured. The paper is improved, however, I still had difficulties with the reporting clarity and think that in many places, less may be more, and suggest some cutting (particularly in the Discussion).

Abstract - I would suggest reordering the first sentences so that the @justinmice sentence comes first as a 'hook' and introduces the paper's main focus - accurate science reporting; then to introduce the content on Alzheimer's disease as essentially the case study you are using to study this (akin to the way you have ordered the Author Summary).

In the Author Summary, perhaps delete "solid" from "solid data" (let the reader judge this) and reword to avoid suggesting causality/directionality (ie state, "science reporting is associated with media reporting").

In the Introduction, I actually liked the ordering the authors had in the earlier version - a paragraph introducing the main focus (scrutiny of science reporting) and I love the 'hook' with @justinmice. Perhaps you can keep the original first paragraph, then introduce the material on Alzheimer's disease and explain explicitly, why this provides such a good case study for studying the phenomenon of omission of the animal model in titles.

It seems the paragraphs beginning line 116 ("This sample) belongs at the beginning of the Results (or could be omitted) and then the section "Our findings" (line 124) in the Discussion or omitted. I would suggest ending the Introduction on line 115.

Results:

The results needs a first paragraph that explains how you arrived at this sample and the two comparison groups. Again, in meta-research papers, a flow digram is often the most expedient way to communicate this, especially to clearly show how many scientific papers in each group resulted in news pieces (with accompanying 'n's). Could you revise Figure 1 to include all aspects of the sampling process including the sampling frame, how many papers were excluded and reasons why.

I had previously raised this, but particularly for descriptive data, including proportions (%) would be meaningful; every proportion should be accompanied by the numerator and denominator so it is really clear to which group or sub-sample you are referring to. It is otherwise very hard to follow. And when providing comparisons, please provide the proportion for each group, e.g "We did find that nondeclarative papers generated significantly more news proportionally (31%) than the papers in the declarative group (X%) (p = 0.012)."

In Figure 1, what is the significant of the last row of "News" - what are the two groups?

For each aspect of the analysis, it would be helpful to have a sub-heading to guide the reader. e.g. to introduce the section beginning "News stories posted online often reproduce the original title of the research paper [15]. . "

Line 239, "constraint" typo; and "mentioning mice or not" and line 240 "limitations"

Discussion:

The Discussion is quite repetitive - I would try to cut it substantially so that the key study findings are highlighted and then briefly placed in the context of the literature. I think with some serious editing, it could be cut by a 1/3 or more and not lose anything.

The first sentence of the discussion 249 is very hard to read; please reword, or just delete. The second sentence makes sense and nicely sums up the main finding.

This sentence seems contradictory: "Interestingly, the The PLoS guidelines on How to Write a Great Title suggests authors to include the organism used in the research, even though none of the PLoS journals asks authors to add the study's organism to the article's title [18]."

Line 311- reword, I am not sure that "overclaims" is a word - perhaps "over-interprets"?

You mention "One example is the guidelines created for The Age, The Sydney Morning Herald, Brisbane Times and WAtoday, in Australia." What do these guidelines actually entail?

The material on 340 reporting post-hoc sub-group analysis should be in the Results and the Discussion of these results separated out.

The material beginning on line 370 is quite speculative and does not really add much. I would suggest deleting this paragraph.

The paragraphs beginning line 397 and 406 should be combined.

---

## [Editor Report · Decision Letter 3]

5 May 2021

Dear Dr Triunfol,

On behalf of my colleagues and the Academic Editor, Lisa Bero, I am pleased to say that we can in principle offer to publish your Meta-Research Article entitled "What's not in the news headlines or titles of Alzheimer’s disease articles? #InMice" in PLOS Biology, provided you address any remaining formatting and reporting issues. These will be detailed in an email that will follow this letter and that you will usually receive within 2-3 business days, during which time no action is required from you. Please note that we will not be able to formally accept your manuscript and schedule it for publication until you have made the required changes.

PRESS

Thank you again for supporting Open Access publishing. We look forward to publishing your paper in PLOS Biology. 

Sincerely, 

Ines

--

Ines Alvarez-Garcia, PhD 

Senior Editor 

PLOS Biology